# Detection and Validation of Macro-Activities in Human Inertial Signals Using Graph Link Prediction

**DOI:** 10.3390/s24041282

**Published:** 2024-02-17

**Authors:** Christoph Wieland, Victor Pankratius

**Affiliations:** 1Bosch Sensortec GmbH, 72770 Reutlingen, Germany; 2Robert Bosch GmbH, 70469 Stuttgart-Feuerbach, Germany; victor.pankratius@bosch.com

**Keywords:** human macro-activity recognition, HAR, activity sequences, activity validation, graph link prediction, graph neural network, GNN

## Abstract

With the continuous development of new wearable devices, sensor-based human activity recognition is enjoying enormous popularity in research and industry. The signals from inertial sensors allow for the detection, classification, and analysis of human activities such as jogging, cycling, or swimming. However, human activity recognition is often limited to basic activities that occur in short, predetermined periods of time (sliding windows). Complex macro-activities, such as multi-step sports exercises or multi-step cooking recipes, are still only considered to a limited extent, while some works have investigated the classification of macro-activities, the automated understanding of how the underlying micro-activities interact remains an open challenge. This study addresses this gap through the application of graph link prediction, a well-known concept in graph theory and graph neural networks (GNNs). To this end, the presented approach transforms micro-activity sequences into micro-activity graphs that are then processed with a GNN. The evaluation on two derived real-world data sets shows that graph link prediction enables the accurate identification of interactions between micro-activities and the precise validation of composite macro-activities based on learned graph embeddings. Furthermore, this work shows that GNNs can benefit from positional encodings in sequence recognition tasks.

## 1. Introduction

Human activity recognition (HAR) with wearable sensors is a growing research field in pervasive computing [1]. For instance, humans can track sports activities [2,3,4] or household activities [5,6] to monitor, support, and improve their daily routines. Published benchmark data sets form an important foundation for the development and implementation of new HAR applications. These data sets contain signals from accelerometers, gyroscopes, or other sensors that are available in wearable devices [4,5,7,8]. Typically, these signals have a single primary activity label. Most published work focuses on improving the classification rates for these activities using increasingly sophisticated deep learning models [9,10].

Besides the recognition of plain human activities, another challenge for sensor-based HAR has emerged in recent years: the detection of *macro-activities* (or *complex activities*) [6,11]. Macro-activities are complex human activities that are composed of various subsequent *micro-activities* (or *meta-activities*) [6,11]. Depending on the use case, micro- and macro-activities can have distinct levels of granularity. Liu et al. refer to the lowest level of movements as *motion units* [12]. Some research projects also investigate the recognition of concurrent or coupled human activities [13].

In table tennis, for example, training exercises (macro-activities) are sequences of strokes (micro-activities) [14]. Furthermore, a single table tennis stroke involves a backswing, a stroke movement, and a swing-out movement (motion units) [15]. The distinction between human micro- and macro-activities, however, is a more general problem that is not particularly restricted to the sports domain. In 2020, a macro-activity recognition competition [16] challenged participants to accurately detect macro-activities and their micro-activities in cooking scenarios. For example, the macro-activity preparing salad consists of various micro-activities, such as taking some vegetables, washing them, slicing them, and putting them in a bowl [6]. Although the distinction between micro- and macro-activities can provide profound information about human behavior [11], such competitions are the exception in state-of-the-art HAR research. A possible explanation for the lack of micro–macro HAR research is the lack of data, as most public HAR data sets only contain either micro- or macro-activity labels [4,5,7,8].

Another challenge in sensor-based HAR is determining when a macro-activity begins and when it ends. Data sets, such as the cooking activity data set [6], typically provide this information by defining time intervals (windows) that represent portions of specific macro-activities. Unfortunately, this information is not directly accessible in real-world scenarios. Therefore, this work explores the adaptation of graph link prediction [17] to detect and facilitate the tracking of valid micro-activity sequences in inertial sensor signals. In this context, link prediction determines whether the next micro-activity can be appended (linked) to the current sequence of micro-activities, thereby forming a valid sequence or marking the beginning of another sequence. This paper builds on graph neural networks (GNNs) [18] because they have previously shown promising results on link prediction tasks [18,19,20]. Additionally, GNNs are underrepresented in the HAR domain, with only a few existing papers on GNN-based HAR (e.g., [21,22,23,24]). This study offers an excellent opportunity to gain a deeper understanding into the application of GNNs within the HAR domain.

This work presents a GNN-based neural network architecture with three components. The components detect and validate micro-activity sequences that form complex macro-activities. In addition, we present an end-to-end approach for integrating a novel macro-activity recognizer into an existing micro-activity recognition workflow. We evaluate our concept on data that we have derived from table tennis exercises [3] and cooking activities [6]. For the table tennis data, we consider training exercises as macro-activities and the corresponding stroke types as micro-activities. For the cooking data, we consider recipes as macro-activities and preparation steps as micro-activities. In both cases, the micro-activities are located within short sliding windows of signal data. The detection of such homogeneous micro-activity windows is enabled by change point detection [25], adaptive sliding windows [26], or joint activity segmentation [27].

This paper has the following structure. Section 2 gives a brief overview of the related work. Section 3 describes our proposed method in detail. Section 4 evaluates our approach on two data sets. Section 5 discusses important considerations regarding the application of our approach. Section 6 provides a conclusion.

## 2. Related Work

Peng et al. [28] presented AROMA, a multi-task learning framework for micro–macro HAR with sensor signals. AROMA uses a CNN to extract task-dependent features from micro-activity windows and an LSTM to classify macro-activities. The LSTM takes fixed-length macro-activity samples that contain a fixed number of subsequent micro-activity CNN embeddings. Peng et al. trained the two structures jointly to capture related features from both tasks.

Hu et al. [29] presented a semi-supervised approach for classifying micro-activities that leverages only a small amount of labeled micro-activity data for training. Since macro-activities implicitly contain information about their underlying micro-activities, the presented framework additionally integrates labeled macro-activity data in the training process to compensate for the missing micro-activity labels. Thus, the framework learns relationships between micro- and macro-activities to perform micro-activity predictions. Since macro-activity labels are easier to obtain and micro-activity labels are limited, the presented approach makes the data preparation for micro–macro HAR much easier.

Xie et al. [11] presented an approach to micro- and macro-activity recognition that is based on template matching. They first take a three-axis acceleration signal and calculate movement angle profiles. Then, they leverage these movement angle profiles to extract separate macro-activity intervals from the signals. Afterward, they split each extracted macro-activity into a sequence of micro-activities and classify it using template matching and dynamic time warping. The resulting sequence of matched templates represents the macro-activity. The authors label the macro-activity using the least edit distance between the classified micro-activity sequence and given macro-activity profiles.

Alia et al. [16] summarized the “Cooking Activity Recognition Challenge”. The task was to classify micro- and macro-activities in three cooking recipes based on predefined windows of accelerometer and motion capture data. Each data set contained 3 s windows with their macro-activity labels (the recipe) and a list of corresponding micro-activities, e.g., take, cut, and put. Several teams have developed a variety of machine learning approaches to solve these two tasks. Mao et al. [30] employed GNNs to classify micro-activities based on skeleton data.

The field of GNNs in sensor-based HAR is still in its infancy, with only a few research papers published. For instance, Wieland et al. [21] transformed sliding windows of signal data into graphs that connect vertices based on temporal relationships between successive time steps. The authors classified such graphs using TinyGraphHAR, a powerful GNN model for sensor-based HAR. Further approaches can be found in [22,23,24].

## 3. Materials and Methods

Our proposed method for detecting and validating macro-activities relies on graph link prediction: the goal is to determine which graph vertices are related [18] and to connect those vertices with edges. We also apply this concept one layer of abstraction up to macro-activity recognition to assess whether successive micro-activities form valid micro-activity sequences.

We now explain important concepts regarding the application of GNNs to micro–macro HAR. This chapter describes the transformation of micro-activity sequences into graph representations (Section 3.1), a novel GNN model architecture (Section 3.2), and the processing steps in our model (Section 3.3).

### 3.1. Mapping Micro-Activity Sequences to Graphs

As a prerequisite for graph link prediction, we need to transform a sequence $M_{T}=\left\{m_{1},\cdots ,m_{t}\right\}$ of micro-activities $m_{t}$ with t∈{1,…,T} into a graph representation $G_{T}=\left(V_{T},A_{T}\right)$ as illustrated in Figure 1.

The micro-activities $m_{t}$ are represented as graph vertices $v_{t}$. Each $v_{t}$ contains an *F*-dimensional micro-activity embedding vector **F** that is either obtained by a micro-activity classifier or a comparable feature extraction technique (we use UMAPs [31] for simplicity). Hence, the set of graph vertices $V_{T}$ has the shape T×F.

The adjacency matrix $A_{T}$ of $G_{T}$ is a square T×T matrix with zeros in all columns except the last one $a\left[:,T\right]={\left[1\right]}^{T}$ (see Table 1). Thus, our proposed micro-activity graph is equivalent to a directed star graph with a self-loop at the central vertex $v_{T}$.

The idea behind this star structure is that the GNN aggregates the individual micro-activity features in its final vertex. Thus, the last vertex becomes a rich representative of the given micro-activity sequence. Furthermore, the sparsity of the graph (and its adjacency matrix $A_{T}$) allows for efficient storage and processing, because only *T* edges exist in the graph. However, this graph representation ignores the temporal information of the micro-activity sequences. To account for the chronological order of micro-activities, we follow the ideas in [32] and enrich the vertex features with positional encodings before we process the graph with our GNN. The utilization of positional encodings streamlines the graph by eliminating the necessity to encode temporal dependencies along the graph edges.

### 3.2. Model Architecture

Our proposed multi-task model consists of three components: the graph embedding block **E**, the graph validator **V**, and the link predictor **L**. Figure 2 illustrates the interplay of **E**, **V**, and **L**. All inner layers leverage the ReLU activation function. The output layers of the binary tasks **V** and **L** use the sigmoid activation function.

The *graph embedding block **E*** forms the input layer to the neural network. It converts a graph $G_{T}=\left(V_{T},A_{T}\right)$ of micro-activities $M_{T}$ into a single embedding vector **E**$\left(G_{T}\right)$. However, the sparse graph structure does not include information about the chronological order of its micro-activities. To compensate for that, we enrich the vertex features with the positional encodings from [32]:(1)$$PE\left(t,f\right)=\left\{\begin{array}{cc} \cos\left(\frac{t}{10,000^{2f/F}}\right) & \mathrm{if}\;f\;\mathrm{is}\;\mathrm{even},\\ \sin\left(\frac{t}{10,000^{2f/F}}\right) & \mathrm{if}\;f\;\mathrm{is}\;\mathrm{odd}, \end{array}\right.$$
where t∈T is the temporal index of a vertex $v_{t}\in V_{T}$ (i.e., a micro-activity $m_{t}\in M_{T}$), *F* is the size of the feature space, and f∈F is the index of a feature. We update the vertex features $V_{T}^{\prime }$ with the element-wise sum of the initial vertex features $V_{T}$ and their positional encodings PE:(2)$$V_{T}^{\prime }=V_{T}⊕PE.$$

Afterward, a single graph attention layer (GAT) [33] with five attention heads and 2F channels aggregates the information of all micro-activities in the last (*T*-th) graph vertex according to their relevance to the macro-activity. Finally, we extract the rich graph embedding **E**$\left(G_{T}\right)$ from the last vertex using a custom Last Vertex Pooling operation.

The *graph validator **V*** checks whether $V_{T}$ represents a *valid* macro-activity or not. If a macro-activity, for example, consists of the three micro-activities *a*, *b*, and *c*, then the only valid embedding is **E**([a,b,c]). Other permutations, e.g., [a], or [a,b], are *invalid*. To make this decision, **V** passes **E**$\left(G_{T}\right)$ through h=3 hidden fully connected layers ${\mathrm{FC}}_{h}^{V}$ with 2F hidden features and a binary output layer ${\mathrm{FC}}_{o}^{V}$.

**Definition 1** (valid and invalid micro activity sequences)**.** *The graph validator **V** returns **1** if the input graph $G_{T}$ represents a **valid** macro-activity, i.e., a micro-activity sequence that represents a complete macro-activity. **V** returns **0** if the input graph $G_{T}$ is an **invalid** macro-activity, i.e., $G_{T}$ is either a subsequence of a macro-activity or not part of a macro-activity.*

The *link predictor **L*** checks whether appending the next micro-activity $m_{T+1}$ to the current micro-activity sequence $M_{T}$ constructs a *possible* micro-activity sequence or not. For this purpose, **L** concatenates the sequence embedding **E**$\left(G_{T}\right)$ and $m_{T+1}$ and passes this new vector through h=3 hidden fully connected layers ${\mathrm{FC}}_{h}^{L}$ with 2F hidden features and a binary output layer ${\mathrm{FC}}_{o}^{L}$. Sticking with the graph validator example, a link is possible if it forms the micro-activity sequences [a,b,c], [a,b], or [a]. In this example, all other permutations of *a*, *b*, and *c* are *impossible*.

**Definition 2** (possible and impossible links)**.**
*The link predictor **L** returns **1** if the concatenation of the input graph $G_{T}$ and the next micro-activity $m_{T+1}$ results in a **possible** micro-activity sequence. **Possible** micro-activity sequences include both partial micro-activity subsequences and complete macro-activities. **L** returns **0** if the concatenation $G_{T}∥m_{T+1}$ is **impossible**, i.e., $m_{T+1}$ cannot be appended to $V_{T}$ as a new vertex $v_{T+1}$.*

### 3.3. End-to-End Classification Pipeline

Given a sequence of T>0 micro-activity embeddings $M_{T}$ and the micro-activity embedding of the next micro-activity $m_{T+1}$, our approach follows the scheme in Figure 3 to identify macro-activities within micro-activity graphs:

Construct a micro-activity graph $G_{T}$ from the micro-activity embeddings $M_{T}$ (cf. Section 3.1).Generate the graph embedding **E**$\left(G_{T}\right)$ using the graph embedding block **E**.Validate **E**$\left(G_{T}\right)$ using the graph validator **V**.Check whether $V_{T}∥m_{T+1}$ is possible using the link predictor **L**.Now, several actions are possible:(a)If **V**(**E**$\left(G_{T}\right)$) is invalid but **L**(**E**$\left(G_{T}\right)$) is possible, set $M_{T+1}=M_{T}∥m_{T+1}$ and go back to (1).(b)If **V**(**E**$\left(G_{T}\right)$) is valid but **L**(**E**$\left(G_{T}\right)$) is impossible, return **E**$\left(G_{T}\right)$.(c)If **V**(**E**$\left(G_{T}\right)$) is valid and if **L**(**E**$\left(G_{T}\right)$) is possible, return **E**$\left(G_{T}\right)$. In addition, set $M_{T+1}=M_{T}∥m_{T+1}$ and go back to (1).(d)If **V**(**E**$\left(G_{T}\right)$) is invalid and if the link is impossible, $G_{T}$ either represents an unknown micro-activity sequence or an interrupted macro-activity. Thus, interrupt the current micro-activity sequence but keep its embedding **E**$\left(G_{T}\right)$ for later use or inspection. Set $M_{T+1}=\left\{m_{T+1}\right\}$ and go back to (1).

## 4. Results

To evaluate our proposed method for identifying and validating macro-activities in wearable sensor signals, we rely on two real-world data sets: the cooking activity data set [6] and the table tennis data set [3]. Section 4.1 introduces the data sets. Section 4.2 and Section 4.3 describe the data preparation steps and the training configuration. Section 4.4 evaluates our approach on the two prepared data sets.

### 4.1. Data Sets, Macro-Activity Labels, and Micro-Activity Labels

The cooking activity data set [6,16] provides accelerometer signals from four subjects during the preparation of three meals (sandwich, salad, and cereals). Four smart devices collected signals at four body locations (smartphone at the right arm and the left hip, smartwatches at the wrists). The four participants had to prepare the dishes using specified ingredients; however, the subjects could choose their preferred order of preparation. In total, the subjects have performed nine distinct micro-activities to prepare the dishes: take, wash, cut, put, mix, open, pour, peel, and *other*. In our work, we omit *other* micro-activities as we cannot assign them to the individual recipes. This results in eight concrete micro-activities.

The cooking activity data set provides 468 snippets of sensor data. Each snippet has a duration of 30 s and represents at least one micro-activity, but the temporal boundaries between the micro-activities within a snippet are unknown. Furthermore, there is no information about the sequential order of the snippets. Thus, we do not know the order of the snippets and to which execution of a recipe a particular snippet belongs. Fortunately, the cooking activity data set description provides general information on which micro-activities are part of each recipe. Based on this information, we have derived seven valid micro-activity sequences (macro-activities): three for sandwiches, three for cereals, and one for fruit salad. The sandwich recipes consist of 11 micro-activities, the cereal recipes contain 9 micro-activities, and the fruit salad recipe has 14 micro-activities. Section A.1 lists the micro-activity sequences per recipe.

The table tennis data set [3] contains human inertial signals of two table tennis players while executing table tennis strokes. The players collected the signals with a Fossil Gen 5E smartwatch at their racket-holding wrist. The data set encompasses 457 time series and 8 different table tennis stroke types: drive, loop, block, push in forehand and backhand. Each time series contains 5 to 15 identical table tennis strokes. Thus, we need additional knowledge on common table tennis stroke sequences to form versatile micro-activity sequences. We gained this knowledge by analyzing common table tennis exercise descriptions from [14]. The exercises and their corresponding stroke sequences define valid micro-activity sequences. We ignore all exercises with stroke types that are not part of the eight stroke types (micro-activities) in the table tennis data set. As a result, we identify 55 unique table tennis exercises (macro-activities). The most complex exercise contains 10 strokes (micro-activities). The shortest valid micro-activity sequence has only one stroke type. Section A.2 lists the micro-activity sequences.

### 4.2. Data Preparation

#### 4.2.1. Micro-Activity Distributions

A prerequisite of our macro-activity recognition GNN is creating a graph structure from raw activity data (cf. Section 3.1). However, the initial data sets lack well-defined micro-activity embeddings that suit the recognition of macro-activities on wearables. We apply the following steps to create representative, multivariate (i.e., *F*-dimensional) distributions to generate random embeddings for each micro-activity class $m_{i}$:Split the signals into non-overlapping windows with 1 s of data. Each window represents one or more micro-activity classes.Fit a seeded UMAP [31] with *F* components on the windows and extract the UMAP embeddings per micro-activity class.For each micro-activity class $m_{i}$, calculate the mean values of their UMAP embeddings $\mu_{i}=\mu \left(m_{i}\right)$ and generate Gaussian micro-activity distributions $N(\mu_{i},\sigma )$ with a pre-defined standard deviation σ.

We repeat this procedure several times with varying random seeds ∈{1,2,3} to account for random statistical behavior. We select small numbers of components F∈{5,15,25} to reduce the computational requirements of the GNN. Furthermore, we investigate different standard deviations σ∈{0.05,0.15,0.25} to evaluate the behavior of the GNN with varying levels of uncertainty.

Table 2 compares the similarities of the micro-activity embeddings per data set. The values represent the average level of similarity across the three random seeds. We define the similarity measure *s* as follows:(3)s=1n2∑i=1n−1∑j=i+1n1d[μi,μj],
where *n* represents the number of distinct micro-activity classes within a given data set and $d\left[i,j\right]=∥\mu_{i}-\mu_{j}∥$ is the Euclidean distance between the mean UMAP embedding vectors of two micro-activities $m_{i}$ and $m_{j}$.

For both data sets, the similarity scores shrink with an increasing number of embedding features *F*. Thus, the embeddings with larger *F*’s are more distinctive.

The micro-activity embeddings of the cooking activity data set exhibit a higher level of similarity compared to those in the table tennis data set because the cooking activity signals often encompass multiple inseparable micro-activities (cf. Section 4.1). Consequently, the sliding windows of the cooking activity data set may carry multiple micro-activity labels that reduce the distinguishability of the UMAP embeddings of different micro-activities.

Figure 4 and Figure 5 illustrate the mean UMAP embeddings $\mu_{i}$ of both data sets for different feature counts *F* in the 2D-plane after applying a two-dimensional principal component analysis (PCA). The diagrams also include circles around the mean vectors representing the standard deviations σ∈{0.05,0.15,0.25}. The plots confirm the conclusions from Table 2, as smaller *F*s exhibit more overlaps and smaller distances between micro-activities.

#### 4.2.2. Sequence Embeddings and Data Split

We have derived seven valid micro-activity sequences for the cooking data set and 55 valid micro-activity sequences for the table tennis data set (cf. Section 4.1). Next, we apply Algorithm 1 on each micro-activity sequence to generate all their possible subsequence embeddings.
**Algorithm 1** Generate possible subsequence embeddings.**Input** $M_{T}$: micro-activity sequenceN(μ,σ): micro-activity distributions**Output** $M_{T}^{\prime }$: embeddings of all possible subsequences of $M_{T}$ that start with $m_{1}$ (the first micro-activity in $M_{T}$)1:$E_{T}\leftarrow \mathrm{empty}\;\mathrm{list}$2:**for** micro-activity class $m_{t}\in M_{T}$ **do**3:    $e_{t}\leftarrow \mathrm{draw}\;\mathrm{embedding}\;\mathrm{for}\;m_{t}\;\mathrm{from}\;N\left(\mu \left(m_{t}\right),\sigma \right)$4:    $E_{T}.append\left(e_{t}\right)$5:**end for**6:$M_{T}^{\prime }\leftarrow \mathrm{empty}\;\mathrm{list}$7:**for** micro-activity embedding $e_{t}\in E_{T}$ **do**8:    $m_{t}^{\prime }\leftarrow E_{T}\left[:t\right]$9:    $M_{T}^{\prime }.append\left(m_{t}^{\prime }\right)$10:**end for**11:**return** $M_{T}^{\prime }$

Algorithm 1 takes a valid micro-activity sequence $M_{T}$ and the micro-activity distributions N(μ,σ) as inputs and returns all possible subsequence embeddings $M_{T}^{\prime }$. For instance, possible subsequences of the table tennis exercise $M_{T}=\{\mathrm{drive},$ loop, push} are $M_{T}^{\prime }=\{\left\{\mathrm{drive}\right\},$ {drive, loop}, {drive, loop, push}}. Thus, the algorithm returns embedding sequences for all three subsequences in $M_{T}^{\prime }$.

We repeat this procedure 30 times for each micro-activity sequence $M_{T}$. We divide the resulting embeddings per micro-activity sequence according to a 60/20/20 rule (60% of the embeddings form the training set, the remaining 40% are evenly distributed between the validation and test sets).

#### 4.2.3. Graphs and Graph Labels

For each subsequence embedding, we create one micro-activity graph according to Section 3.1. Each of these graphs requires two labels: the graph validator label and the link predictor label.

Graphs representing valid micro-activity sequences, i.e., complete macro-activities, receive the graph validator label *1*. Graphs representing incomplete subsequences receive the graph validator label *0*.

For the link prediction task, our GNN requires additional link embeddings $m_{T+1}$ as inputs. For each graph $G_{T}$, we extract $n_{\mathrm{pos}}$ possible successor embeddings $m_{T+1}$ from the next longer graphs $G_{T+1}$, where $G_{T+1}=G_{T}∥m_{T+1}$. These possible links receive the link predictor label *1*. In addition, we randomly draw $n_{\mathrm{imp}}$ impossible successor embeddings from the previously generated embeddings:(4)$$n_{\mathrm{imp}}=\left\{\begin{array}{cc} 0.1*n_{\mathrm{total}} & \mathrm{if}\;n_{\mathrm{pos}}=0,\\ \bar{n_{\mathrm{pos}}} & \mathrm{else}, \end{array}\right.$$
where $n_{\mathrm{total}}$ is the total number of previously generated micro-activity embeddings and $\bar{n_{\mathrm{pos}}}$ is the average count of possible link embeddings per micro-activity. $n_{\mathrm{pos}}=0$ if a graph represents a valid micro-activity sequence that has no possible subsequent micro-activities. Impossible links receive the link predictor label *0*.

This results in 284,916 graphs for the cooking activity data and 853,062 samples for the table tennis data. Table 3 lists the label distributions of the graph validator and the link predictor in the train splits of both data sets.

Due to the following reasons, the labels are highly imbalanced. For each complete micro-activity sequence $M_{T}$, we derive *T* subsequences with different lengths. Only the *T*-th sequence represents the complete macro-activity. Thus, most graphs represent incomplete, invalid micro-activity sequences. Moreover, we generate impossible link predictor samples for all micro-activity classes that cannot follow a subsequence based on our derived macro-activities. Especially for longer micro-activity sequences, there are only a few possible subsequent micro-activity classes. Therefore, the number of impossible link predictor samples grows quickly.

### 4.3. Model Training

Our multi-output model learns two loosely coupled tasks in parallel. The model updates the weights of **V** and **L** based on their respective binary cross-entropy losses and the weights of **E** with respect to the summed losses of all components. We balance the samples per class and task to ensure that unequal class distributions do not disturb the training. We use the Adam optimizer [34] with a learning rate of 0.001 for optimization and a batch size of 128. To reduce overfitting, we apply early stopping with patience 10 and a maximum number of 500 epochs.

### 4.4. Quantitative Analysis

We evaluate our GNN models using various data configurations. We alter the micro-activity embedding sizes *F*, vary σ during data generation, and investigate the effects of positional encodings on the graphs. To obtain robust results, we repeat each experiment three times with varying random seeds (1, 2, 3). Figure 6 presents the mean accuracies and the mean macro F1 scores for the cooking activity data and Figure 7 for the table tennis data. The plots also contain the standard deviations of the metrics.

#### 4.4.1. Cooking Data (Figure 6)

The performances of **V** and **L** increase with a growing number of features *F*. We expect this behavior since adding more features to the UMAP embeddings reduces the similarity between the support vectors per micro-activity class (cf. Table 2). Thus, it improves the distinguishability of the samples.

Shrinking σ also improves the performances since a smaller σ results in narrower Gaussian distributions for data generation. Thus, it increases the inner similarities of the micro-activity classes.

Furthermore, adding positional encodings PE improves graph validation and link prediction across all experiments. Again, we expect this behavior since our graph transformation completely ignores the temporal relationships between subsequent micro-activities (cf. Figure 1). The differences between the addition and the omission of PE increase with growing σ and are most noticeable in the macro F1 scores of the graph validator **V**. Even with small σ=0.05, the F1 scores of the graph validator without PE are up to 22.2 percentage points worse than with PE. Moreover, adding PE enables **V** to accurately validate graphs with an F1 score of up to 96.2%, even for the largest σ=0.25. These results highlight the importance of positional encodings for the reliability of the models and their potential to improve GNN-based HAR models.

Another interesting finding is that the graph validator outperforms the link predictor across all settings. The differences between both components become larger with growing σ. Thus, the link prediction task requires more precisely distinguishable micro-activity embeddings than graph validation. Nevertheless, **L** yields remarkable F1 scores of more than 89 for F≥15 features and σ≤0.15 thanks to the assistance of positional encodings.

#### 4.4.2. Table Tennis Data (Figure 7)

The general observations from the cooking data set also hold for the table tennis data set. The accuracies and the F1 scores exhibit consistent improvements with growing *F*. However, due to more pronounced variations among individual micro-activity classes in the table tennis data set, the differences in metric scores across diverse *F* dimensions are notably smaller compared to the cooking activity data set.

Furthermore, shrinking σ has a similar effect in the table tennis scenario as for the cooking data set, but in general, the impact is smaller. This observation is especially evident for the link predictor. But, even with σ=0.25, **L** reaches F1 scores greater than 93% for F≥15 features.

The performance metrics of the table tennis data also benefit from positional encodings. For instance, PE improves the macro F1 score of **V** by up to 28.7 percentage points and the F1 score of **L** by up to 25.7 percentage points for σ=0.05. The impact of PE slightly shrinks with growing *F*. Nevertheless, the improvements from adding PE remain well above 15 percentage points for both model components. Again, these findings underline the importance of positional encodings in graph-based sequence recognition tasks but also emphasize the importance of a distinguishable feature space.

Similar to the cooking activity data set, the graph validator outperforms the link predictor in the table tennis scenario. However, the differences are less pronounced because **L** performs better overall, especially with σ≤15. With a difference of 3 percentage points, we observe the most significant gap for F=5 and σ=0.25. With F=5 features, the differences shrink to less than 0.6 percentage points. Hence, even with the largest σ=0.25, the gap between **V** and **L** becomes negligible, provided that the micro-activities are easily distinguishable.

## 5. Discussion

The unique graph structure eliminates the temporal component from the micro-activity sequences by directly connecting all micro-activities only to the last micro-activity in time. To overcome this limitation, we enrich the vertex features with positional encodings. This enhancement preserves the essential temporal information within the micro-activity graphs. Our evaluation underscores the effectiveness of this method, highlighting its ability to accurately verify micro-activity graphs.

One drawback of our approach is that it alters the requirements for data preparation compared to conventional sensor-based HAR scenarios. Instead of labeling windows solely with a single activity class, our method necessitates additional information for detecting and validating micro-activity sequences. Specifically, this includes knowledge about possible micro-activity sequences (i.e., which activity follows another). To our knowledge, sensor-based HAR data sets with high-quality micro- and macro-activity labels are currently unavailable. Consequently, we had to derive our evaluation data from two existing HAR data sets with limited details on the interplay between micro- and macro-activities.

Our data curation strategy involves the generation of both possible and impossible micro-activity sequences. This procedure rapidly inflates our micro–macro-activity data sets, particularly in scenarios where there is either a large or a small number of distinct micro-activity classes that can follow the current micro-activity sequence (many possible links or many impossible links). This issue emphasizes the need for future sensor-based HAR data sets to include detailed meta-information about micro- and macro-activities and their sequential order. Future data sets with continuous sensor signals and clearly defined activities will no longer require the generation of synthetic micro-activity sequences. Instead, the sequence information provides labels for possible links, while we only need to generate impossible links as negative samples for training the link predictor. Thus, such data sets will streamline the development of robust micro–macro HAR models. Moreover, our proposed solution to graph-based link prediction for detecting macro-activity sequences in sensor signals requires unambiguous micro-activity windows that serve as graph vertices. Current research suggests methods such as change point detection [25], adaptive sliding windows [26], or joint activity segmentation [27] to generate such homogeneous windows.

As described in Section 3.3 Step (5d), our proposal allows for the detection and storage of invalid, impossible micro-activity sequences. Storing incomplete sequences raises the question of how to deal with interrupted micro-activity sequences. One potential solution involves reloading previously interrupted micro-activity sequences after completing the subsequent macro-activity instead of initializing a new activity sequence. Depending on the number of previously interrupted micro-activity sequences, however, this approach may slow down the overall classification pipeline. Thus, developing a retention policy becomes crucial to manage memory usage and computational costs.

An alternative to the GNN-based prediction of possible links between micro-activities are hidden Markov models (HMMs). According to [35], HMMs are applicable to (sensor-based) HAR. However, detecting and validating human macro-activities with HMMs requires extensive exploration of how to transform micro-activities and their varying feature embedding vectors into HMM states and observations. Solving this challenge will enable a future comparison between GNN-based and HMM-based link prediction approaches.

## 6. Conclusions

In this work, we have explored the application of GNNs for predicting links between subsequent human micro-activities. We have presented a model architecture that leverages the power of GNNs to predict links between micro-activities and previously executed micro-activity sequences. This way, link prediction surpasses the limitations of basic activity classifications and opens up plenty of new possibilities for pervasive HAR applications. For instance, this approach enables athletes to carefully track their training exercises, including the validation of performed exercises or the suggestion of follow-up activities. This information allows players to optimize their executed activity sequences in a goal-oriented manner.

Furthermore, we have observed that positional encodings positively effect GNNs for sensor-based macro-activity recognition. Positional encodings eliminate the need to model the complex temporal relationships of sensor signals in graphs. In the future, we want to explore the potential benefits of combining positional encodings with graph neural networks or convolutional neural networks to enhance traditional sensor-based HAR tasks.

## Figures and Tables

**Figure 1 sensors-24-01282-f001:**
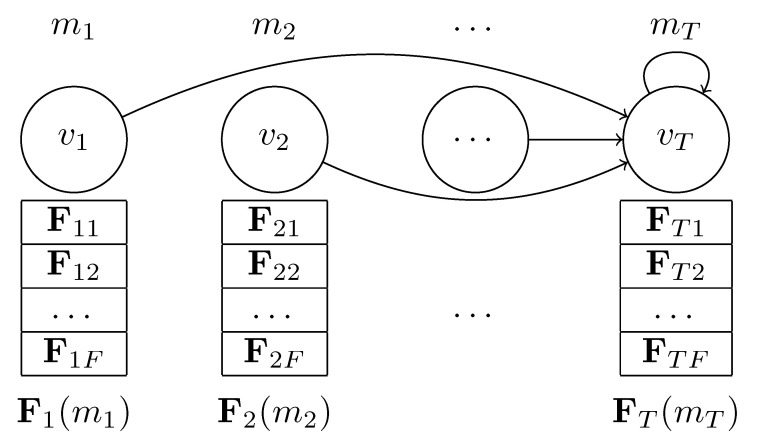
The graph $G_{T}$ of a micro-activity sequence $M_{T}$. The micro-activities $m_{t}\in M_{T}$ represent vertices $v_{t}\in V_{T}$ with *F*-dimensional feature embedding vectors **F**. Each $v_{t}$ has one outgoing edge that connects it to the last vertex $v_{T}$.

**Figure 2 sensors-24-01282-f002:**
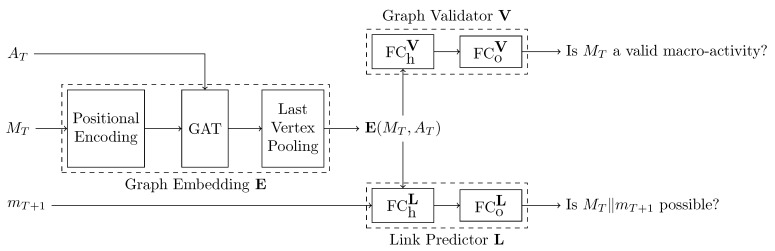
Our macro HAR GNN consists of the graph embedding block that calculates an embedding **E** of a micro-activity graph $G_{T}=\left(V_{T},A_{T}\right)$ using a graph attention layer (GAT), the graph validator that checks whether **E**$\left(G_{T}\right)$ represents a valid macro-activity, and the link predictor that checks whether $M_{T}∥m_{T+1}$ forms a possible micro-activity sequence, where $m_{T+1}$ is the next macro-activity. ${\mathrm{FC}}_{h}$: *h* fully connected hidden layers; ${\mathrm{FC}}_{o}$: fully connected output layers.

**Figure 3 sensors-24-01282-f003:**
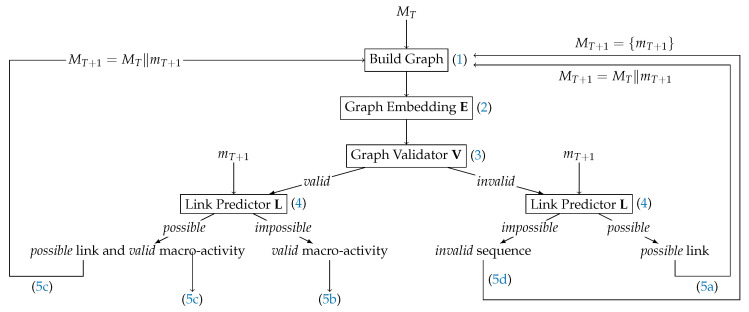
End-to-end classification pipeline. Inputs: initial micro-activity sequence $M_{T}$ with T>0 micro-activities and the next micro-activity embedding $m_{T+1}$. The terms *valid* and *invalid* are defined in Definition 1, and the terms *possible* and *impossible* are explained in Definition 2.

**Figure 4 sensors-24-01282-f004:**
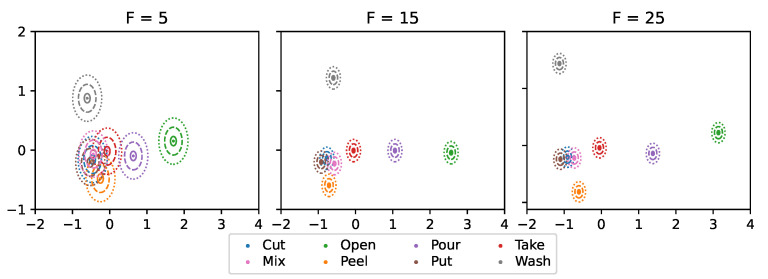
Two-dimensional PCA plot of the mean UMAP embeddings (seed 1) of the cooking activity data set (dots: centroids, solid/ dashed/ dotted circles: standard deviations σ∈{0.05,0.15,0.25}).

**Figure 5 sensors-24-01282-f005:**
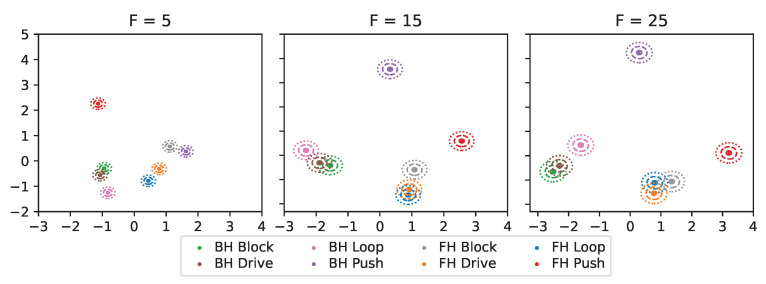
Two-dimensional PCA plot of the mean UMAP embeddings (seed 1) of the table tennis data set (dots: centroids, solid/ dashed/ dotted circles: standard deviations σ∈{0.05,0.15,0.25}).

**Figure 6 sensors-24-01282-f006:**
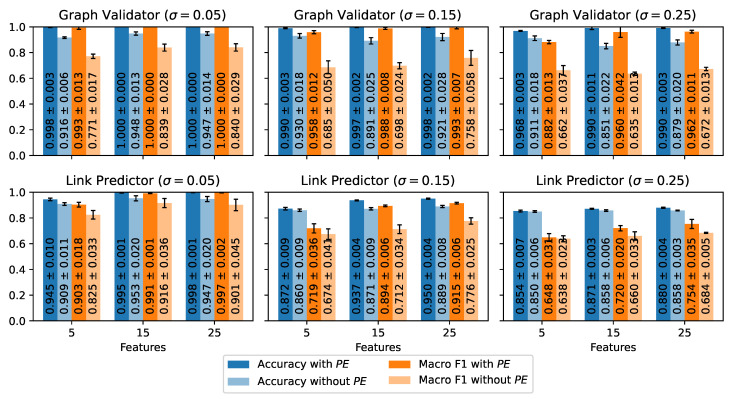
Test results of the graph validator and the link predictor on the cooking data set. The plots illustrate the mean accuracies and the mean macro F1 scores for different standard deviations σ and varying micro-activity embedding sizes *F*. Each plot compares the results with and without the positional embedding PE.

**Figure 7 sensors-24-01282-f007:**
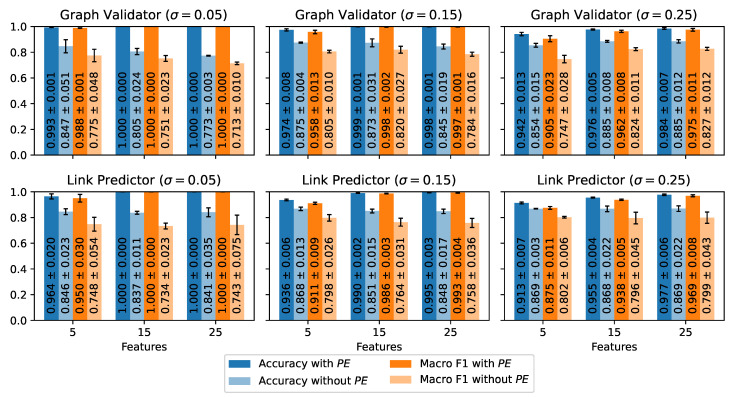
Test results of the graph validator and the link predictor on the table tennis data set. The plots illustrate the mean accuracies and the mean macro F1 scores for different standard deviations σ and varying micro-activity embedding sizes *F*. Each plot compares the results with and without the positional embedding PE.

**Table 1 sensors-24-01282-t001:** The sparse adjacency matrix $A_{T}$ of $G_{T}$. The last column contains 1s. The other cells contain zeros.

$A_{T}$	$v_{1}$	...	$v_{T-1}$	$v_{T}$
$v_{1}$	0	.	0	1
$v_{2}$	0	.	0	1
...	.	.	.	.
$v_{T}$	0	.	0	1

**Table 2 sensors-24-01282-t002:** Average similarities of the micro-activity embeddings per data set. Lower values indicate better distinguishability between micro-activities.

Data Set	Features *F*		
	5	15	25
Cooking Activity	1.7235	1.0429	0.8657
Table Tennis	0.5071	0.3024	0.2322

**Table 3 sensors-24-01282-t003:** Label distributions of the graph validator and the link predictor in the training splits of the cooking activity data set and the table tennis data set.

Data Set	Graph Validator		Link Predictor	
	Valid	Invalid	Possible	Impossible
Cooking Activity	16,254	217,134	41,796	191,592
Table Tennis	134,784	564,066	174,960	523,890

## Data Availability

The cooking activity data set is openly available in [6]. The table tennis data set [3] is closed.

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
