# Peer review of "Detection and Validation of Macro-Activities in Human Inertial Signals Using Graph Link Prediction"

_sensors, 2024, doi:10.3390/s24041282_

Round 1

Reviewer 1 Report

Comments and Suggestions for Authors

This work tried to explore the advancement of sensor-based human activity recognition. Authors employed graph link prediction and Graph Neural Networks to identify interactions between micro activities, which had certain impact of positional encodings in sequence recognition tasks. However, the organization structure of this manuscript is quite chaos. The quality of the figures presented in the manuscript is poor and insufficient, which cannot reach up the standard of the journal Sensors. Therefore, this paper is not recommended for publication in current form. Some specific comment listed below may be helpful for improving the quality of this work.

1. Authors are suggested to reorganize the structure of the manuscript, and make it coherence and clear.

2. The language of this manuscript needs to be polished.

3. The study should provide more details on the real-world datasets used for evaluation, including their size, diversity, and source.

4. The limitations and potential challenges associated with the application of graph link predication in this work should be provided.

Comments on the Quality of English Language

Extensive editing of English language required

Author Response

Dear Reviewer 1,

thank you for your feedback. We have carefully read your suggestions and made the following changes to the manuscript to improve its quality:

  1. We added more details regarding the data set characteristics, such as size and number of subjects. We also added a short Appendix that describes the micro-activity sequences that we used in the evaluation.
  2. Furthermore, we discussed more challenges and limitations of GNN-based micro-macro HAR in Section 5 – Discussion. We also improved the existing discussion by better highlighting the challenges already mentioned.
  3. We also improved the language of the manuscript. For example, we converted British English words into American English, changed passive voice to active voice, fixed grammatical errors, and simplified long sentences.

However, we were not able to understand all of your suggestions and kindly ask you to elaborate on them:

  1. Authors are suggested to reorganize the structure of the manuscript, and make it coherence and clear.”
    We have followed the official MDPI Sensors guidelines for the manuscript sections (Sensors | Instructions for Authors | Research Manuscript Sections). The introduction introduces the topic, the related work section discusses prior publications, materials and methods explains the core contribution, including the signal-to-graph transformation and the GNN architecture, the results section evaluates our proposal, and the paper closes with the discussion and the conclusion. The only difference to the official guidelines is the added related work section. However, since MDPI Sensors allows free-format submissions, this should not be an issue. In our opinion, the only debatable question is whether the data set description fits better in Section 3 - Materials and Methods or in Section 4 – Results. However, as the data is not the core contribution of our work, this description is better placed in the evaluation section.
  2. The quality of the figures presented in the manuscript is poor and insufficient, which cannot reach up the standard of the journal Sensors.”
    Figures 1 to 3 illustrate the graph structure, the GNN architecture, and the classification pipeline of our proposed solution. We have created these figures using tikz in Latex. The figures show good readability at different scales and are well structured.
    Figures 4 and 5 are 2D PCA-plots. We have created these figures using matplotlib in Python. Again, the figures show good readability at different scales and the legend is readable.
    Figures 6 and 7 are vertical bar charts. Each bar contains the bar value with its standard deviation. Again, we have created these figures using matplotlib in Python. The figures show good readability at different scales. The legend and the bar labels are readable.

Best regards,

The authors of “Detection and Validation of Macro-Activities in Human Inertial Signals using Graph Link Prediction

Reviewer 2 Report

Comments and Suggestions for Authors

Nice job.

Admittedly, unlike many mature pattern recognition fields, such as speech recognition and image recognition, many definitions in HAR have not been unified. But I still have some considerations.

I believe that for wearable HAR, the academic community has some other definitions beyond "macro/micro/meta/several subsequent micro" activities, which tend to be more comprehensive in various scenarios: single motion versus motion sequence of CONCURRENT, COUPLED, and/or SEQUENTIAL motions (doi.org/10.26092/elib/1219, Sec. 1.1). I think the macro/meta/several subsequent definitions ignore at least concurrent and coupled activities (sitting and watching TV; driving; walking and answering the phone; cutting food...). In addition, according to the generalized sequence model of human activities (motion units), micro activity is not the smallest unit of activity that can be recognized, but motion units (e.g., preparing, flight, and landing phases in jumping).

When you mentioned the lack of micro-macro labels in datasets nowadays, I suggest you discuss the latest feature-based information retrieval of multimodal biosignals with a self-similarity matrix, which can black box-wise segment and label HAR data. A large sliding window can partition a macro activity of walking/position movement automatically, while a small sliding window can subdivide the macro one into "walking upstairs", "walking straight forward", and "walking downstairs", which you call "micro". In addition, I also recommend exploring Yale Hartmann's latest landmark work on HAR high-level design. These features can be micro/macro independent.

I noticed that your model for mapping micro activities is very similar to hidden Markov models. In fact, I would consider it to be reverse complete HMMs or, in another way, forward HMMs split into several sub-models (starting with v1, v2, ... respectively). Can you make a comparison and analysis of this? HMM has been widely studied in HAR, both for micro activities and "macro" activities (such as HHMM). E.g., doi.org/10.1007/978-981-19-0390-8_108. Thanks to its inherent excellent modeling performance for time series (this is also reflected in your mapping graph), HMM has even been reported to have an accuracy that exceeds deep learning in many works. Moreover, HMM also has advantages that DL does not have, such as training efficiency, interpretability, expandability, and generalization.

Pay attention to some grammatical errors based on differences in language sense, such as:

THESE information (I guess you translated diese Informationen?)

And the following "information ALLOW"

only few research papers exist -> only A few

You can use British English or American English, but you shouldn't mix them in the same manuscript. For example, labelled and labeled appear one after another.

Comments on the Quality of English Language

See above.

Author Response

Dear Reviewer 2,

thank you for your thorough feedback. We have carefully read your suggestions and made the following changes to the manuscript to improve its quality:

  1. We added a reference to motion units (10.23919/EUSIPCO54536.2021.9616298) in the second paragraph of the introduction and also added a concrete example of motion units in the third paragraph. Furthermore, we added a note regarding concurrent and coupled activities to capture the broad context. Here, we use the original reference (10.5445/IR/1000049328) that was mentioned in Section 1.1 of the proposed source, because the provided reference does not consider these kinds of activities.
  2. As you have pointed out correctly, a large sliding window can span multiple micro-activities, while a small sliding window has a narrower view. To highlight challenges regarding the selection of a proper window size, we added references to change point detection, adaptive sliding windows, and joint activity segmentation to the second-last paragraph of the introduction. These techniques help to find homogenous micro-activity windows in time series data.
  3. We also improved the language of the manuscript. For example, we converted British English words into American English, changed passive voice to active voice, fixed grammatical errors, and simplified long sentences.
  4. We did not compare our approach with your HMM proposal. This comparison would go beyond the scope of this paper. Instead, we discussed HMM as an alternative approach in the last paragraph of the discussion and included the proposed reference (doi.org/10.1007/978-981-19-0390-8_108).

Best regards,

The authors of “Detection and Validation of Macro-Activities in Human Inertial Signals using Graph Link Prediction

Round 2

Reviewer 1 Report

Comments and Suggestions for Authors

Some modifications have been made to enhance the quality of the manuscript. In the submitted manuscript, figure 1 and figure 2 is missing. The revised article could be considered for publication after correcting the issues.

Comments on the Quality of English Language

Language is acceptable.

Author Response

Dear Reviewer 1,

thank you for your response. We added the missing figures.

Best regards
The authors of “Detection and Validation of Macro-Activities in Human Inertial Signals using Graph Link Prediction

Reviewer 2 Report

Comments and Suggestions for Authors

I argue to accept this manuscript.

Author Response

Dear Reviewer 2,

thank you for your positive feedback.

Best regards,
The authors of “Detection and Validation of Macro-Activities in Human Inertial Signals using Graph Link Prediction